# Prevalence and Antibiogram Pattern of *Acinetobacter baumannii* from 2013 to 2023 in a Tertiary Care Hospital in the Western Region of Saudi Arabia

**DOI:** 10.3390/antibiotics14030274

**Published:** 2025-03-07

**Authors:** Ohood Alharbi, Hamdi M. Al-Said, Sami S. Ashgar, Naif A. Jalal, Hani Faidah, Aiman M. Momenah, Ayman K. Johargy, Farkad Bantun, Fadi S. Qashqari, Omar Hani Faidah, Mamdouh A. Bukhari, Karem Ibrahem

**Affiliations:** 1Department of Microbiology and Parasitology, Faculty of Medicine, Umm Al-Qura University, Makkah 21955, Saudi Arabia; osharbi@uqu.edu.sa (O.A.); ssashgar@uqu.edu.sa (S.S.A.); najalal@uqu.edu.sa (N.A.J.); hsfaidah@uqu.edu.sa (H.F.); ammomenah@uqu.edu.sa (A.M.M.); akjohargy@uqu.edu.sa (A.K.J.); fmbantun@uqu.edu.sa (F.B.); fsqashqari@uqu.edu.sa (F.S.Q.); 2Faculty of Medicine, Umm Al-Qura University, Makkah 24382, Saudi Arabia; s439007417@uqu.edu.sa; 3Regional Laboratory, General Directorate of Health Affairs, Makkah City, Ministry of Health, Makkah 24321, Saudi Arabia; mamdouhb@moh.gov.sa; 4Department of Clinical Microbiology and Immunology, Faculty of Medicine, King Abdulaziz University, Jeddah 21589, Saudi Arabia; kaibrahem@kau.edu.sa

**Keywords:** *Acinetobacter baumannii*, MDR, Saudi Arabia, carbapenem resistance

## Abstract

*Acinetobacter baumannii* is pathogen of global concern. It causes infection, especially among immunocompromised individuals in intensive care units, due to its ability to survive for long periods on hard surfaces and under a wide range of environmental conditions and become resistant to almost all the available antibiotics used in clinical practice. **Objectives**: This study aims to address the gap in *A. baumannii* surveillance in Saudi Arabia by tracking the prevalence, patterns, and trends of acquired *A. baumannii* resistance at a healthcare facility in the western part of Saudi Arabia over eleven years. **Methods**: The study was conducted in a tertiary care hospital in the western region of Saudi Arabia, from January 2013 to December 2023. **Results:** Our data indicated that *A. baumannii* infections were predominantly observed in inpatients admitted to the hospital (96%) compared to those treated as outpatients in the emergency clinic (4%). The mean of annual *A. baumannii* infections isolated from inpatients is 503.3, whereas the mean for outpatients is 21, indicating a statistically significant difference with a *p*-value of <0.0001. The analysis of the antimicrobial susceptibility profile of *A. baumannii* demonstrated a variable levels of resistance to the evaluated antibiotics. The lowest resistance rate was for colistin. **Conclusions**: In conclusion, the incidence patterns of *A. baumannii* isolates peaked in 2013, then declined, and have recently shown an increase, underscoring the necessity for proactive interventions to curtail its dissemination, notwithstanding initial decreases in infection rates and resistance.

## 1. Introduction

*Acinetobacter baumannii* (*A. baumannii*) is an opportunistic Gram-negative pathogenic bacterium, often associated with aquatic environments. Also, it has high infection rates among immunocompromised individuals, especially patients treated for long periods in healthcare facilities [1]. *A. baumannii* has become a major cause of infection in hospitals, especially intensive care units (ICUs), because this pathogen can survive for long periods of up to several months on hard surfaces and under a wide range of environmental conditions, making it a major problem facing infection control committees in many hospitals [2,3,4,5,6]. Direct contact between patients infected with *A. baumannii* and healthcare providers, and indirect contact with contaminated instruments in the hospital and healthcare settings, are among the main routes of transmission and spread of *A. baumannii* infection, especially pneumonia [7,8]. Several infections associated with this bacterium have been reported, such as ventilator-associated pneumonia, wound infections, meningitis, bloodstream infections, and urinary tract infections, confirming its role as a major cause of hospital-acquired infections [9,10,11,12]. The most serious infections caused by *A. baumannii* are ventilator-associated pneumonia and bacteremia, which are common among patients in ICUs, with mortality rates reaching more than 40% [13]. Many reports of outbreaks have been published worldwide, but we note that reports from high-income countries are more numerous than those from low-income countries, and this could be due to the lack of sufficient resources to detect pathogens [14]. During the COVID-19 pandemic, *A. baumannii* infections have been reported to cause resistant co-infections [15]. *A. baumannii* strains possess virulence factors such as outer membrane protein A (OmpA), lipopolysaccharide (LPS), and sophisticated secretion mechanisms, such as type II and VI secretion systems, in addition to physical factors such as surface hydrophobicity, which help them form biofilms, cause infections in different parts of the human body, and evade killing by antibiotics, making them resistant to many antibiotics [16,17,18,19,20,21,22]. Therefore, antibiotic-resistant *A. baumannii* strains emerged in the early 1990s, spread to the Middle East in 2006, and became a serious concern by 2015 in the European Union [23,24]. The emergence of multidrug-resistant (MDR) *A. baumannii* strains, especially CRAB, in healthcare facilities and hospitals has become a major burden and challenge in the management and treatment of infections caused by these MDR strains and has become an emerging issue worldwide [25,26,27]. Studies indicate that rates of MDR *A. baumannii* are increasing in many parts of the world and in Southeast Asia and the Arab region [28]. Consequently, the presence of MDR *A. baumannii* in many hospitals in Saudi Arabia has become a major health and economic issue [29,30]. As a result, these strains have become devastating pathogens in hospitals, potentially leading to a pandemic [31]. Several studies have investigated the factors leading to the spread of MDR *A. baumannii* infections worldwide [32]. A study of ICUs found that the prevalence of *A. baumannii* infection was 5.6% in Western Europe, 17.1% in Eastern Europe, 3.7% in North America, 13.8% in Central and South America, 4.4% in Oceania, 19.2% in Asia, and 14.8% in Africa [33]. Contaminated ventilators, mechanical ventilation, prolonged hospital stay, contaminated hands of healthcare workers, and prolonged broad-spectrum antibiotic therapy have been implicated in the spread of MDR *A. baumannii* infections [34,35]. Due to the development of resistance mechanisms in *A. baumannii* strains, this has affected the effectiveness of carbapenems against *A. baumannii* strains, which has become a major challenge in treating infections of these strains [36,37]. This increasing trend of concurrent antibiotic resistance has raised great concern worldwide, especially in many European countries.

In 2017, resistance rates reached more than 50% in several European countries, with 60% in Poland, 75% in Italy, and 80% in Greece, resulting in high mortality rates ranging from 30% to 75% [38]. Consequently, multiple strains of *A. baumannii* resistant to antibiotics, especially third-generation cephalosporins and carbapenems, have emerged as a major public health threat in hospitals and healthcare settings [38]. Studies indicate high mortality rates from *A. baumannii* infection associated with hospital-acquired pneumonia and ventilator-associated pneumonia, ranging from 22% to 49% in the United States, while the rates in Southern Europe were (55.7%), Western Asia (56.2%), and North Africa (53.3%). In the Mediterranean region, countries such as Egypt (53.3%), Turkey (61.4%), and Greece (68.2%) reported the highest death rates [39]. In Germany, studies indicated that infection with *A. baumannii* is widespread in many federal states [36]. Countries such as Belgium and the United Kingdom report lower rates of antibiotic resistance [35,38]. A recent study conducted in Saudi Arabia revealed the prevalence of *A. baumannii* infection in several Saudi hospitals. The high rates of multidrug-resistant *A. baumannii* indicate the need for intensive research on the implications of increased risk of multidrug-resistant *A. baumannii* infection [3]. Another study conducted at King Abdulaziz Hospital in Saudi Arabia indicated that MDR *A. baumannii* increased from 55% in 2010 to 67% in 2013 [40]. Nonetheless, there exists a significant gap in the study of trends regarding *A. baumannii* infections and the alterations in antimicrobial resistance among these this critical pathogen infections. Accordingly, this study aimed to monitor the prevalence and pattern of resistance of acquired *A. baumannii* in a healthcare hospital in the western region of Saudi Arabia over eleven years.

## 2. Result

### 2.1. Distribution and Rate of Isolation

This retrospective study aimed to assess the burden of infections attributable to *A. baumannii* across a decade, from January 2013 to December 2023. Of the 55,102 isolates identified during the study period, 5777 (11%) were *A. baumannii*. The annual counts varying from 768 to 340, with an average of 525 per year. The annual count trend of *A. baumannii* specimens (Figure 1 top-line) indicates that the peak was observed in 2013, with 12% (768) instances recorded. In 2016, the number of cases decreased by 2% (468). The decline in cases continued to improve until 2021, with the lowest yearly incidence recorded in 2018, totaling merely 340 *A. baumannii* cases. In 2022 and 2023, the cases increased to 13% (540) and 11% (570), respectively.

Infection/colonization with *A. baumannii* was more commonly observed among inpatients admitted in the hospital (96%) than outpatients treated in the ER or clinic (4%). The mean of *A. baumannii* isolated from inpatients is 503.3, whereas the mean for outpatients is 21.7, indicating a statistically significant difference with a *p*-value of <0.0001. Table 1 shows the department-wise distribution of clinical samples. The rate of *A.baumannii* isolation (Figure 1) was highest among patients treated in the ICU (51%), followed by patients in medical wards (19%) and the surgical ward (17%). *A.baumannii* isolation rate from burn unit, ER, and pediatric ward accounted for total of 11%. Table 1 presents the annual case count for each department.

As seen in Table 2, during the entire study period, isolations of *A. baumannii* were most frequently identified from the sputum samples (2636; 46%), followed by wounds (1564; 27%) and blood (873; 15%). Other samples, including urine, catheter tip, and miscellaneous samples, constitute less than 10% each; the total counts and percentage for each sample is seen in Table 2. Irrespective of the sample type, *A. baumannii* isolate percentages were elevated in inpatient specimens.

### 2.2. Patient Demographics

Table 3 presents the gender distribution of patients infected with *A. baumannii*. A total of 67% of *A. baumannii* samples were collected from male patients (3917), while female patients accounted for 33% of the total, with 1861 patients. In regard to age, patients in the age group of 61–75 exhibit a markedly elevated infection rate relative to other age groups (*p* < 0.0001) as seen in Figure 2. The difference in the mean number of *A. baumannii* between the 61–75 age group and the 46–60 age group is significant (*p* = 0.04). To evaluate potential inherent factors influencing the prevalence and frequency of cases over time, the data was visualized in a Sankey diagram (Appendix A). The frequency of *A. baumannii* isolation appears to be stable over the eleven-year duration. Unlike female patients, who often contract *A. baumannii* at the age of 61 or older, male patients are predominantly affected at a younger age. *A. baumannii* is more frequently isolated from blood samples in patients younger than 61, whereas wounds diagnosed with *A. baumannii* are more commonly found in patients older than 61. Samples of sputum, urine, and others were collected uniformly across all age groups.

Appendix A and the original data in the Appendix A suggest that a screening was established at 2016, revealing the incidence of *A. baumannii* in screened samples mostly taken from the ICU patients. The yearly quantity of inspected samples fluctuated considerably. The number of positive samples varied from a minimum of 5 in 2021 to a maximum of 21 in 2016 and 2017.

### 2.3. Antibiotic Susceptibility Profile of A. baumannii

The analysis of the antimicrobial susceptibility pattern of *A. baumannii* revealed variable changes in the resistance rate in most of the tested antibiotics (Table 4, Figure 3A–C). Throughout the duration of the study, the highest resistance rate was observed for beta-lactam antibiotics, including ampicillin, piperacillin/tazobactam, and cefotaxime at 100%, whilst colistin had the lowest resistance rate at 3.7% (Table 4). The tested beta-lactam antibiotics, including penicillin, cephalosporins, and carbapenems, showed an alarming high rate of resistance over the eleven years, with a gradual increase in the resistance rate (Table 4 and Figure 3A). Similarly, ciprofloxacin exhibited a constant high resistance rate over the eleven years. In contrast, TMP/SMX and levofloxacin showed a fluctuation in the reported resistance rate, followed by sharp decrease in 2021 (Table 4 and Figure 3B). Similar but lesser reductions were observed in aminoglycoside antibiotic (gentamicin, and amikacin). Notably, the resistance rate to colistin fluctuated over the eleven years but with no statistically significant difference between 2013 and 2023.

Upon analyzing the resistance rates from the beginning and the final two years of the study, it was observed that the years 2013 and 2014 exhibited significantly higher rates compared to the last two years, 2022 and 2023, for levofloxacin, TMP/SMX, and aminoglycosides. However, this was not the case for the other antibiotics such as beta-lactams, colistin, and ciprofloxacin, as illustrated in Figure 3D.

### 2.4. Prevalence of Carbapenem-Resistant A. baumannii

The evaluation of carbapenem-resistant *A. baumannii* (CRAB) isolate rates and distribution across departments and patient age over the 10 years was conducted by calculating the isolates reported as resistant to both imipenem and meropenem or identified as positive for carbapenem resistance. Figure 4 illustrates a decrease in both the rate and total number of CRAB, dropping from 70% (511 isolates) in 2013 to 60% (340 isolates) in 2023. In 2018, we were surprised to find that the reported lowest impedance corresponded with the highest percentage of CRAB infections. The COVID-19 pandemic did not affect the total number or rate of CRAB infections during 2020 and 2021. The data indicates that the ICUs exhibit the highest incidence of CRAB infections, with reported cases increasing from 21 in 2013 to 150 in 2023.

## 3. Discussion

*Acinetobacter baumannii* is an opportunistic pathogen that imposes a significant burden on healthcare systems, necessitating challenging treatment due to the evolution and dissemination of antimicrobial-resistant strains. In 2024, the WHO designated the carbapenem-resistant *A. baumannii* strain as a critical pathogen, acknowledged for its considerable threat to public health due to limited treatment options and significant disease burden [41]. Ongoing and proactive monitoring of *A. baumannii* is essential for the implementation of effective infection control strategies and for assessing the initiatives aimed at curbing the dissemination of this pathogen. Nonetheless, in Saudi Arabia, there have been limited investigations carried out to observe the trends of antimicrobial resistance in *A. baumannii*. Therefore, to address the existing gap in surveillance and present an updated overview of the antibiotic resistance status for this significant pathogen, our study examines the prevalence and trends of antibiotic resistance over a decade, specifically from 2013 to 2023, within a tertiary hospital in Makkah, one of the largest populated cities in Saudi Arabia that hosts millions of pilgrims annually.

*A. baumannii* exhibits a prevalence rate of 11%, identified in 5777 out of a total of 54,663 clinical samples. This rate is consistent with the most recent prevalence reported in other regions of the kingdom [3,42]. The annual count of *A. baumannii* specimens indicates that the peak was in 2013 with 768 cases, while the lowest occurred in 2018 with 340 samples. The number and rate of samples showed a decline to 468 samples (10%) in 2016, and the decrease in cases continued to fall until 2021, reaching 475 (8%). However, in 2022 and 2023, the cases rose to 540 (13%) and 570 (11%), respectively. The decline may be attributed to the implementation of infection control measures aimed at reducing the transmission of *A. baumannii* infections. As the data indicates, between 2016 and 2023, the hospital recorded results for *A. baumannii* screening (Appendix A), which could explain the substantial reduction in positive rates from 14% prior to 10% or lower. Consistent with the overall drop in *A. baumannii* case numbers and rates, the resistance rates also dropped. This decline may also have been influenced by the establishment of the “Health Facilities Infection Control General Department” organization, which was tasked with tackling the issues of AMR and hospital-acquired infections, and the implementation of the Gulf Cooperation Council States’ strategic plan to combat antimicrobial resistance in 2015 [43].

Antibiotic antibiogram data showed a constant high prevalence of resistance throughout the study years. The resistance rate for ampicillin and piperacillin/tazobactam in penicillin has been found to be 95% on average. The resistance rates for third- and fourth-generation cephalosporins were found to be 96% for ceftazidime and 97% for both cefotaxime and cefepime. The prevalence of carbapenem resistance in *A. baumannii* isolates ranges from 60% to 70%. A similar trend was found in *A. baumannii* antibiotic resistance. This percentage aligns with the reported prevalence of CRAB infections in other regions of Saudi Arabia [44] and worldwide [45].

Over the course of the years observed, there has been a notable decline in the rate of resistance in *A. baumannii* infections. The most significant drop was noted in levofloxacin resistance, which decreased from 80% in 2013 to 43% in 2023. Similarly, cotrimoxazole (trimethoprim-sulfamethoxazole) resistance fell from 91% to 54%. The reduction in resistance to levofloxacin and TMX may be attributed to the prescription policy restrictions established by the Ministry of Health in 2018 and during the COVID-19 pandemic in 2020 [46], since the resistance percentage notably diminished in 2020. Comparable results have been documented globally [47,48,49]. Nevertheless, without knowledge of the antibiotic usage and prescription practices or the genotypes of the circulating strains at this hospital, we cannot ascertain the precise reasons for this drop. Similarly, gentamicin resistance percentage also dropped from 75% in 2013 to 60% in 2023, amikacin from 81% to 74%, and CRAB infections from 70% to 60%. A high antibiotic resistance rate of 67% or higher, such as in ceftazidime, cefepime, imipenem, meropenem, gentamicin, ciprofloxacin, and trimethoprim/sulfamethoxazole, was reported [50].

Colistin is the “last line” of therapy for *A. baumannii* infections that are resistant to almost all other antibiotics. In our study, colistin resistance is averaged at 13%. The yearly prevalence rate of colistin-resistant isolates was below 20%, except for 42% which occurred in 2014. The increase in colistin resistance percentage in 2014 noted for *P. aeruginosa* [51] might be related to the laboratory methods of testing or hospital outbreak of colistin-resistant strains due colistin overuse or inadequate infection control measures. There are notable differences in the frequency of colistin-resistant infections between studies. In Asia, however, the prevalence of colistin resistance varies from 4% in China to 50% in the United Arab Emirates [52]. Healthcare practitioners are forced to use more colistin due to the current high level of resistance to all other antibiotic classes and the dearth of novel antimicrobial medications, which is a major contributing cause to the rise in colistin resistance [53].

Patient demographics indicate that males are significantly more affected by *A. baumannii* infections. However, it is important to note that the majority of total isolates were recovered from male patients, accounting for 60% of the cases. Although this percent of increased male infection with MDR strains is in accordance with previous studies for other pathogens [3,54], a thorough investigation of other clinical confounders and comorbidities is needed to determine with accuracy why males are more affected. Patients aged 61–75 were significantly more affected than any other age group, as with older age the immune system function declines. However, other studies proved that *A. baumannii* is more prevalent in older age.

The demographic analysis of *A. baumannii*-positive samples indicated that sputum constituted the greatest proportion, accounting for 47% of the total samples. This indicates that *A. baumannii* is associated with respiratory tract infections. According to prior research from 2005 to 2009, Acinetobacter species were responsible for 26.5% of ventilator-associated pneumonia caused by B-lactamase Gram-negative producers in Saudi Arabia [28]. Wound samples are the second most prevalent, with 27% of all samples. Infections caused by *A. baumannii* are linked to severe injuries and fractures that need external fixation [55], which are frequently treated in this hospital.

ICUs are recognized hotspots for healthcare-acquired infections, such as Acinetobacter species, because of their high frequency of invasive procedures and susceptible patients with serious comorbidities [56]. Furthermore, ICU patients on mechanical ventilation had a tenfold higher risk of *A. baumannii* infection [57]. In line with epidemiological studies that have showed a higher isolation rate of *A. baumannii* colonization and epidemics among ICU patients than any other hospital departments [58]. Our findings reveal that more than 50% of the 5777 total cases were isolated from ICU patients. Another study showed the prevalence of *A. baumannii* and CRAB colonization of the gastrointestinal tract in intensive care unit (ICU) patients in Saudi hospitals was 8% and 6%, respectively [59]. Furthermore, studies showed that ICU admission will increase the chance for *A. baumannii* infections and patients with *A. baumannii* infections experienced a considerably longer ICU stay [60].

This study encompasses eleven years of data from the largest public hospital in Makkah; its specificity to a distinct geographical and healthcare context may restrict the applicability of its findings to other places in Saudi Arabia. Nevertheless, the data analysis in this study assist in understanding the variations in antibiotic resistance levels over a decade. Consequently, comprehending antimicrobial resistance requires careful monitoring of data concerning resistance prevalence and drug utilization. The Saudi Ministry of Health has advocated for antimicrobial stewardship initiatives as an essential strategy for combating antibiotic resistance, proving success in decreasing resistance rates. The findings may provide insights into antibiotic resistance development and can be utilized to assess the efficacy of antimicrobial stewardship programs in shaping prescribing practices. The high rates of *A. baumannii* infections in the ICU department necessitates improving hygiene and infection control measures. Sputum is the predominant sample type for *A. baumannii*, necessitating particular attention to the ventilation system. The COVID-19 pandemic has positively influenced the dissemination of multidrug-resistant organisms. The adoption of universal measures against infectious diseases, including social distancing, travel reduction, and enhanced personal hygiene, substantially restricts the transmission of pathogens and multidrug-resistant infections.

While this study provides significant insights into the prevalence and resistance patterns of *A. baumannii* over an extended period, it has limitations, including reliance on retrospective data and the lack of morbidity and mortality statistics for *A. baumannii* infections, which limits the generalizability of the findings. Also, this study lacks genomic surveillance of the strains, which would effectively determine the strains and inform infection prevention and control strategies by delineating outbreaks and identifying sources and pathways of bacterial transmission. Hospital studies have shown the effectiveness of whole-genome sequencing (WGS) in the high-resolution characterization of *A. baumannii* outbreaks or persistent populations [61,62]. The study was conducted at a single center, potentially limiting its ability to represent broader epidemiological trends. Future research must prioritize multicenter trials and incorporate comprehensive genomic analysis to enhance understanding of resistance mechanisms and formulate more effective treatment strategies. Also, research must prioritize the development of innovative antimicrobial agents and alternative therapeutic strategies to combat multidrug-resistant *Acinetobacter baumannii*. Additionally, analyzing the impact of comprehensive antibiotic stewardship programs across various healthcare settings is crucial for identifying best practices and mitigating the spread of resistance. Continuous genomic surveillance can provide significant insights into the evolutionary pathways of resistance and inform more targeted interventions.

In conclusion, this research aims to fill a knowledge gap about antibiotic resistance of *A. baumannii* infections in Saudi Arabia. The study conducted at Makkah Tertiary Hospital involved a retrospective analysis of eleven years of samples. The findings highlight the persistent threat of this opportunistic bacteria, especially in view of its developing resistance to antibiotics. The highest prevalence of *A. baumannii* was present in sputum and wound clinical samples. The incidence patterns of *A. baumannii* isolates peaked in 2013, then declined, and have recently shown an increase, underscoring the necessity for proactive interventions to curtail its dissemination, notwithstanding initial decreases in infection rates and resistance. Notably, the demographic information highlights a higher prevalence among older patients and males, especially in intensive care units. Importantly, we suggest more research in this area with more cities and centers, as well as more information on prescription trends and patient attributes including mortality and comorbidity.

## 4. Materials and Method

### 4.1. Study Setting

The current study was conducted in a tertiary care hospital in the western region of Saudi Arabia between January 2013 and December 2023 to assess antibiotic susceptibility patterns among *A. baumannii* isolates as a retrospective observational study. The demographic, microbiological, antimicrobial treatment, and patient outcome data were gathered from the data sheets of the patients using a record review approach to analyze *A. baumannii* data from the microbiology database in a tertiary care facility. In this investigation, 55,102 patients were included in the study. Patient data and identity in this study were concealed, including age, year of injury, or any personal data. Therefore, individual patient consent was not required according to ethical guidelines on research data.

### 4.2. Bacterial Isolates and Antibiotic Susceptibility

Samples were collected from different wards, i.e., intensive care units (ICUs), medical wards, the surgical ward, burn unit, emergency room admissions (ERAs), and pediatrics, and included different sample types, such as sputum, wound, blood, urine, catheter tip, and others (miscellaneous less than 10% each). Standard microbiology techniques were performed. All data were collected in accordance with ethical standards throughout the study. Microbial identification was conducted according to standard laboratory procedures. Microbial identification of the *A. baumannii* isolates patient samples were cultured on various agar media, including blood agar (Oxoid, Hampshire, UK, CM0259) and MacConkey agar plates (Oxoid, CM0007), and placed in a standard incubator at 35–37 °C for 18–24 h. Aerobic and anaerobic blood culture bottles were processed at the Microbiology Laboratory using the Biomerieux Bact Alert Virtuo automated instrument (bioMérieux Inc., Durham, NC, USA) for blood culture, incubating until a positive signal. Following the incubation period, *A. baumannii* colonies were identified using the Vitek-2 (bioMérieux, Marcy-l’Étoile, France) automated system (GN-21341 cards were used for identification, whereas the N291, N292, and N204 cards were used for antibiotic susceptibility) and Microscan walkaway automated system (Negative Breakpoint Combo 50, NBC 50, Beckman Coulter, Brea, CA, USA) following the manufacturer’s instructions [39]. This study evaluated the following antibiotics: piperacillin/tazobactam, ceftazidime, cefepime, cefotaxime, imipenem, meropenem, gentamicin, amikacin, ciprofloxacin, levofloxacin, and TMP/SMX. The interpretation of the MIC data was based on Clinical Laboratory Standards Institute (CLSI) criteria [63].

### 4.3. Statistics and Data Analysis

A database was created to store and sort important data, such as the total number of patients, specimen type, hospital wards, and antibiograms. The antibiotic susceptibilities were recorded as a percentage of resistance for all *A. baumannii* isolates. Chi-square tests were carried out to compare the susceptibility and the antibiotic class, from different years. Furthermore, T-tests and ANOVA were employed for evaluating mean differences between two groups, such as gender and hospitalization status, as well as among categories of more than two groups, including age groups, sample types, and hospital departments. A *p*-value of 0.05 or less was determined to be statistically significant for all tests. Finally, the statistical analysis was performed with GraphPad Prism version 10.

## Figures and Tables

**Figure 1 antibiotics-14-00274-f001:**
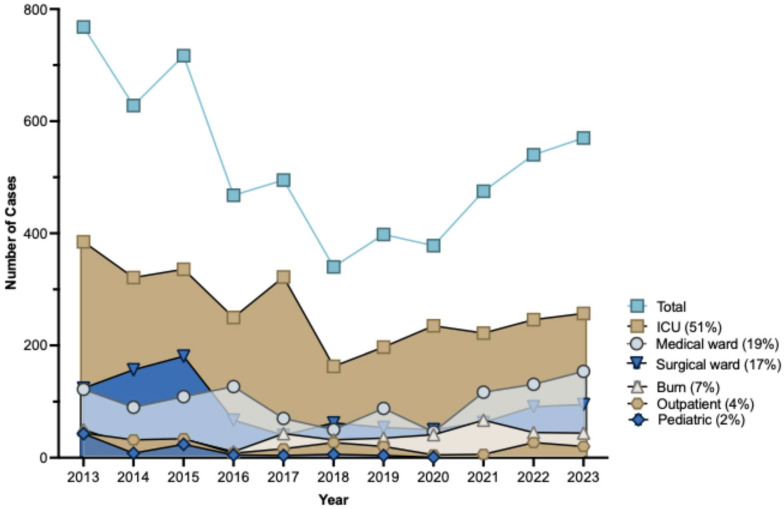
The annual trend of *A. baumannii* cases. The uppermost line illustrates the yearly total of *A. baumannii* cases, peaking in 2013 (768 cases) before subsequently seeing a slow fall to its lowest incidence in 2018 (340 cases), followed by a modest annual increase in total infections by 2023 (570 cases). The bottom lines represent annual cases in each of the hospital departments. The ICU department has the highest incidence of *A. baumannii* infections, accounting for over fifty percent of total cases.

**Figure 2 antibiotics-14-00274-f002:**
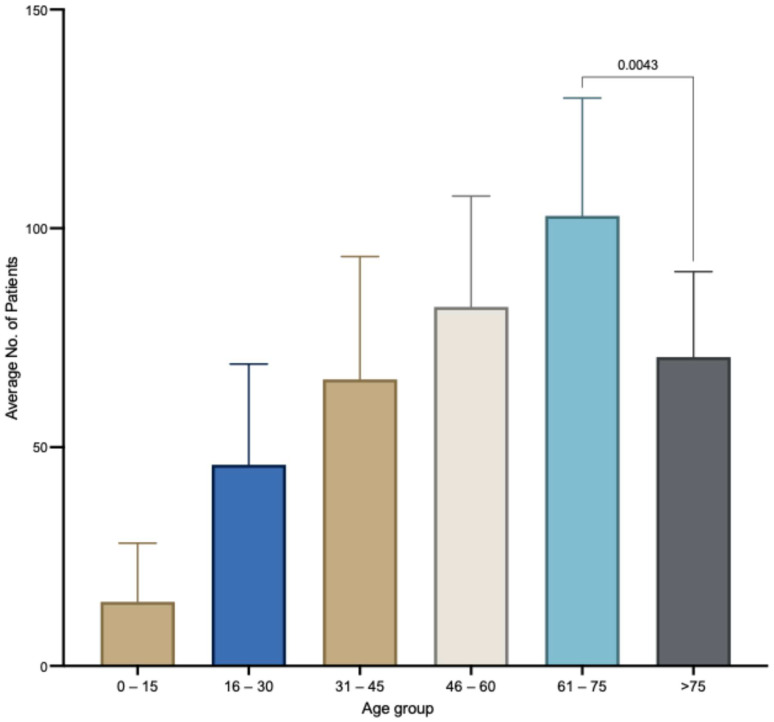
The average age of patients infected or colonized with *A. baumannii.* The age group of 61–75 years was statistically significantly more affected than all other age groups. The age group of 46–60 years was the second most affected by *A. baumannii* infection. The comparison among groups was conducted using a standard one-way ANOVA. The error bar indicates the variability in number of patients for each age group overall years of the study.

**Figure 3 antibiotics-14-00274-f003:**
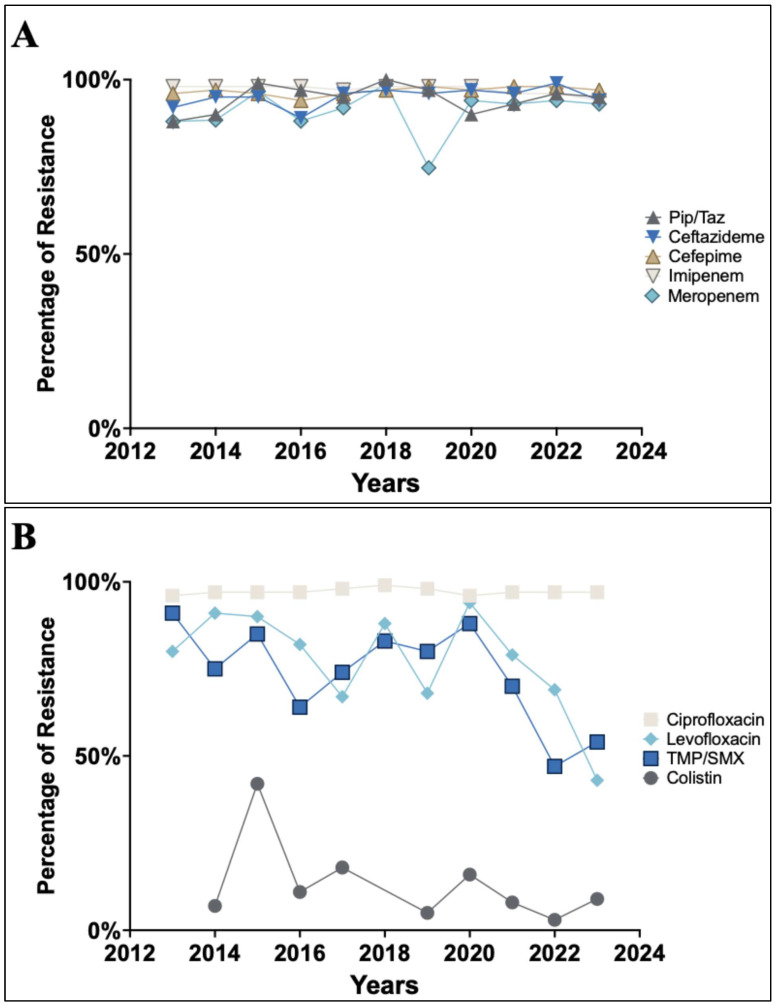
Antimicrobial resistance trends of *A. baumannii*. (**A**) Resistance trend to ß-lactam antibiotics. (**B**) Resistance trend to aminoglycoside. (**C**) Resistance trend to colistin, levofloxacin, ciprofloxacin and cotrimoxazole (trimethoprim-sulfamethoxazole). (**D**) Analysis of the average variations in resistance for amikacin, aminoglycosides, levofloxacin, and TMP/SMX antibiotics observed between the initial and final two years of the study. A statistically significant reduction in resistance rates to aminoglycosides, levofloxacin, and TMP/SMX has been observed. The error bars in (**D**) represent the variation in the initial or final two years. Note: ns means non-significant difference noted from the *p* value.

**Figure 4 antibiotics-14-00274-f004:**
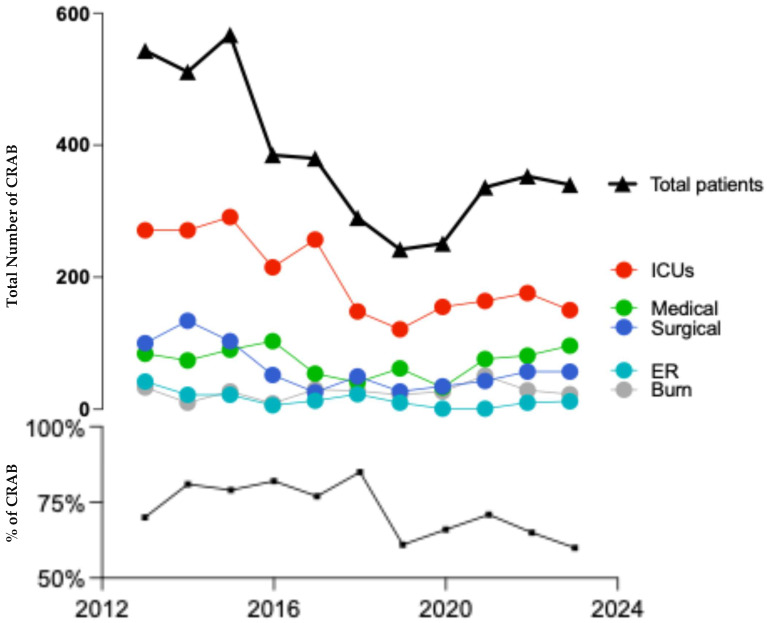
The rate and number of CRAB infections across the 10 years. The bottom black line shows the percentage of CRAB strains (*Y*-axis) over the 10 years. The top part of the figures shows the exact counts of CRAB isolates in total (black line) and in each hospital department. The *Y*-axis in the top part of the figure ranges from 0 to 600 numbers of isolates.

**Table 1 antibiotics-14-00274-t001:** Sample demographics according to the word and admission status. The percentage adjacent to the number represents the ward’s proportion of total cases for that year. The isolation of *A. baumannii* is more likely in hospitalized patients.

Years	Inpatient					Outpatient	Total
	Burn	Pediatric	Medical Wards	Surgical Wards	ICUs	ER and Clinic	
2013	50 (6.5%)	43 (5.9%)	122 (16.9%)	124 (17.1%)	385 (53.2%)	44	768
2014	20 (3.4%)	8 (1.3%)	90 (15.1%)	157 (26.3%)	321 (53.9%)	32	628
2015	33 (4.8%)	24 (3.5%)	109 (16.0%)	181 (26.5%)	336 (49.2%)	34	717
2016	11 (2.4%)	5 (1.1%)	127 (27.6%)	67 (14.6%)	250 (54.3%)	8	468
2017	43 (9.0%)	4 (0.8%)	70 (14.6%)	40 (8.4%)	322 (67.2%)	16	495
2018	32 (10.2%)	6 (1.9%)	50 (16.0%)	62 (19.8%)	163 (52.1%)	27	340
2019	35 (9.3%)	4 (1.1%)	88 (23.3%)	54 (14.3%)	197 (52.1%)	20	398
2020	41 (11.0%)	1 (0.3%)	46 (12.3%)	50 (13.4%)	235 (63%)	5	378
2021	67 (14.3%)	-	117 (24.9%)	63 (13.4%)	222 (47.3%)	6	475
2022	45 (8.8%)	-	131 (25.5%)	91 (17.7%)	246 (48.0%)	27	540
2023	44 (8.0%)	-	154 (28.0%)	95 (17.3%)	257 (46.7%)	20	570
Total	421	95	1104	984	2934	239	5777

**Table 2 antibiotics-14-00274-t002:** Distribution of *A. baumannii* isolates by sample origin and year. *A. baumannii* represents 11% of all isolates collected over a period of eleven years, totaling 5777 positive isolates. Sputum is the most common type of sample affected by *A. baumannii*.

Sample Type, Quantity, and %		Year											Total
		2013	2014	2015	2016	2017	2018	2019	2020	2021	2022	2023	
Sputum	*A.* *baumannii*	382	312	313	229	227	143	175	171	181	254	249	2636
	Total isolates	1417	1173	1086	912	778	842	785	803	897	789	1079	10,561
	% of *A. baumannii*	27%	27%	29%	25%	29%	17%	22%	21%	20%	32%	23%	25%
Blood	*A.* *baumannii*	78	66	89	68	107	38	61	76	100	80	110	873
	Total isolates	1075	382	1015	972	1083	1083	1208	1075	2038	1200	1130	12,261
	% of *A. baumannii*	7%	17%	9%	7%	10%	4%	5%	7%	5%	7%	10%	7%
Catheter tip	*A.* *baumannii*	16	15	4	2	4	4	4	3	7	3	7	69
	Total isolates	117	78	58	76	67	54	45	40	65	41	37	678
	% of *A. baumannii*	14%	19%	7%	3%	6%	7%	9%	8%	11%	7%	19%	10%
Wound	*A.* *baumannii*	172	167	224	120	129	117	128	100	147	129	131	1564
	Total isolates	1563	1560	1442	1274	1164	1164	1136	881	1183	826	1150	13,343
	% of *A. baumannii*	11%	11%	16%	9%	11%	10%	11%	11%	12%	16%	11%	12%
Urine culture	*A.* *baumannii*	26	11	23	11	10	12	3	6	16	24	15	157
	Total isolates	1050	523	674	774	918	918	1136	565	948	593	786	8885
	% of *A. baumannii*	2%	2%	3%	1%	1%	1%	0%	1%	2%	4%	2%	2%
Others	*A.* *baumannii*	94	57	64	38	18	26	27	22	24	50	58	478
	Total isolates	874	812	759	714	917	917	769	665	959	625	924	8935
	% of *A. baumannii*	11%	7%	8%	5%	2%	3%	4%	3%	3%	8%	6%	5%
Total	*A.* *baumannii*	768	628	717	468	495	340	398	378	475	540	570	5777
	Total isolates	6096	4528	5034	4722	4927	4978	5079	4029	6090	4074	5106	54,663
	% of *A. baumannii*	13%	14%	14%	10%	10%	7%	8%	9%	8%	13%	11%	11%

**Table 3 antibiotics-14-00274-t003:** Total count of isolated *A. baumannii* and the gender demographics of the sampled population. The *A. baumannii* isolates were predominantly found in male patients, with a *p*-value of <0.0001.

Year	*A. baumannii* Isolates	Male	(%)	Female	(%)
2013	768	538	70	230	30
2014	628	447	71	181	29
2015	717	474	66	243	34
2016	468	330	70	138	30
2017	495	326	66	169	34
2018	340	212	62	128	38
2019	398	247	62	151	38
2020	378	253	67	125	33
2021	475	319	67	156	33
2022	540	378	70	162	30
2023	570	393	68	177	32
Total	5777	3917	67	1860	33

**Table 4 antibiotics-14-00274-t004:** The percentage of resistance based on antibiogram results. The table indicates the rate of resistance in isolated *A. baumannii*. The resistance rate to the beta-lactam class ranges between 88% and 100%. The resistance rate for the aminoglycoside class ranges between 31% and 95%. The resistance rate to quinolone antibiotics varies between 43% and 99%. The resistance rate to colistin ranges between 3% and 42%. The resistance to TMP/SMX varies between 47% and 91%. Note: NED means Not Enough Data.

Antibiotic	Year-Wise Prevalence (%) of Resistant *A. baumannii*										
	2013	2014	2015	2016	2017	2018	2019	2020	2021	2022	2023
Ampicillin	98.8	NED	100%	100%	100%	88.2	99	100	NED	73.3	99.4
Piperacillin/Tazobactam	88.2	90.0	100	97.4%	95.9	100	97.0	90.0	93.9	96.3	95.0
Ceftazidime	92.0	95.9	95.6	89.9%	96.2	97.5	96.8	97.7	96.8	99.1	94.1
Cefotaxime	96.2	96.7	98.8	96.1%	96.5	100	100	89.5	100	100	NED
Cefepime	96.5	97.3	96.7	94.5%	96.6	97.5	98.6	97.4	98.0	98.4	97.9
Imipenem	98.2	98.9	98.3	97.4%	96.2	98.7	98.6	98.6	96.9	96.0	94.6
Meropenem	88.4	96.6	88.1	91.9%	91.0	98.7	74.7	94.4	93.2	94.9	93.8
Gentamicin	75.6	90.7	88.3	81.2	78.7	59	48.5	63.2	47.0	45.6	60.7
Amikacin	81.4	92.4	31.4	94.9	90.2	84.6	84.5	89.3	69.0	53.8	74.8
Ciprofloxacin	96.4	97.2	97.1	97.1	98.7	99.3	98.3	96.5	97.6	97.5	97.3
Levofloxacin	80.2	91.8	90.8	82.0	67.6	88.4	68.9	94.6	79.7	69.0	43.4
Colistin	NED	7.1	42.9	11.3	18.3	NED	5.3	16.5	8.3	3.7	9.3
TMP/SMX	91.0	75.7	85.9	64.5	74.3	83.0	80.0	88.8	70.8	47.0	54.0

## Data Availability

The original contributions presented in this study are included in the article/Appendix A. Further inquiries can be directed to the corresponding author.

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
