# Peer review of "Prevalence and Antibiogram Pattern of Acinetobacter baumannii from 2013 to 2023 in a Tertiary Care Hospital in the Western Region of Saudi Arabia"

_antibiotics, 2025, doi:10.3390/antibiotics14030274_

Round 1
Reviewer 1 Report
Comments and Suggestions for Authors
This article introduces the drug resistance data of Acinetobacter baumannii in a hospital in Saudi Arabia, which lasted for 10 years and is quite helfpul for this researching field. However, tthe data synthesis and visualization are not satisfied, which requires further exploration and in-depth analysis.
1. do a horizontal comparison with other similar studies in literature, e.g., add some data of mortality and incidence of complications in infected patients, which are often reported in other documents.
2. Use Sangi diagram and data exploration tools to visualize the contribution of different sources of infection, especially, to analyze connections between the flux and possible intrinsic correlations.
3. Use mathematical model or machine learning tools to predict and analyze the time-series data, so as to explore possible influencing mechanisms inside the data.
4. Investigate the other related facts of 2016, such as antibiotic usage and epidemiological data, to analyze the other possible reasons for the decrease and subsequent increase in infection rates in 2016.
5. Ethical commitee conclusions regarding the data disclosure in this paper, if possible.
6. Original data can be listed and included in supplementary materials, so analysis, explanations and conclusions can be highlighted in the main text.
Author Response
For research article
|
Response to Reviewer X Comments
|
||||||||||||||||||||
|
|
||||||||||||||||||||
|
|
|
|
||||||||||||||||||
|
(Reviewer 1)
Manuscript number: antibiotics-3436078 Manuscript title: [Prevalence and Antibiogram Pattern of Acinetobacter baumannii from 2013 to 2023 in a Tertiary Care Hospital in the Western Region of Saudi Arabia].
Dear Reviewer, thank you very much for taking the time to review this manuscript. Please find the detailed responses below and the corresponding revisions/corrections highlighted/ in the re-submitted files.
|
||||||||||||||||||||
Reviewer 2 Report
Comments and Suggestions for Authors The study presents how has been the state of the art of infections caused bu Acinetobacter baumanni reports during one decade ( 2013-2023) The study suggests a sensitive decrease in the most recent years in comparison with the first years of the study. The major prevalence was among the hospitalized patients. The data is tdetailed with very clear graphics and tables. The data presented and discused properly is related to wet region of Saudi Arabic and presents alfso important detailed data about antibiotic resistance of the isolates . All this epidemiological data can help the scientific and medical community to design strategies of control of infections that is off concern, specially in patients submitted to treatments in hospitals. This study analyses the incidence of Acinetobacter baumannii infections and resistance i impatients and outpatients in Saudi Arabia. The data is analysesd and discuted properly and offer impotant data for epidemiologycal studies and infection control strategies.
Author Response
Author's Reply to the Review Report (Reviewer2)
Manuscript number: antibiotics-3436078
Manuscript title: [Prevalence and Antibiogram Pattern of Acinetobacter baumannii from 2013 to 2023 in a Tertiary Care Hospital in the Western Region of Saudi Arabia].
Dear Reviewer, thank you very much for taking the time to review this manuscript.
Reviewer 3 Report
Comments and Suggestions for Authors
Minor Comments:
- Improve the resolution of Figure 1A and Figure 3 for better readability.
- Address the cutoff issue on the letter "C" in Figure 3.
Major Comments:
- Please clarify what each color represents in Figure 1A, either in the caption or in the main text.
- In Section 4.3, authors need to provide confidence intervals or additional statistical measures to strengthen the interpretation of significance. Ensure clarity in explaining how statistical methods, such as the chi-square test, were applied.
- Expand the discussion to address potential reasons for specific antibiotic resistance trends, particularly the decline in resistance to levofloxacin and TMP/SMX. Relate these trends to changes in local prescribing practices or infection control measures.
- Provide more interpretation in the discussion regarding the outlier year 2014 for colistin resistance. Discuss whether this reflects a methodological anomaly or a genuine clinical concern.
- While the study confirms established trends, such as high prevalence in ICUs, male predominance, sputum as the primary sample, the authors should highlight unique findings or propose interventions tailored to Saudi Arabia to add new values.
Author Response
Author's Reply to the Review Report (Reviewer 3)
Manuscript number: antibiotics-3436078
Manuscript title: [Prevalence and Antibiogram Pattern of Acinetobacter baumannii from 2013 to 2023 in a Tertiary Care Hospital in the Western Region of Saudi Arabia].
Dear Reviewer, thank you very much for taking the time to review this manuscript. Please find the detailed responses below and the corresponding revisions/corrections highlighted/in track changes in the re-submitted files.
|
Section |
Reviewer comment |
Author response
|
|
#- Major Comments:
|
1. Please clarify what each color represents in Figure 1A, either in the caption or in the main text. |
We are grateful for your recommendation. Figure 1A displays the total number of cases in each year. The year was represented by the colors, which were created using Graphpad Prism 10. Revision: the figure was redesigned to enhance quality and eliminate ambiguity. |
|
2. In Section 4.3, authors need to provide confidence intervals or additional statistical measures to strengthen the interpretation of significance. Ensure clarity in explaining how statistical methods, such as the chi-square test, were applied.
|
We appreciate your suggestion to enhance the manuscript. |
|
|
3. Expand the discussion to address potential reasons for specific antibiotic resistance trends, particularly the decline in resistance to levofloxacin and TMP/SMX. Relate these trends to changes in local prescribing practices or infection control measures. |
Thank you for addressing this point, The potential factors for TMP resistance drop were discussed. Line 60 -66 in discussion section. |
|
Discuss whether this reflects a methodological anomaly or a genuine clinical concern. |
In relation to this invaluable suggestion. We would like to clarify that the resistance rate for colistin was 42% in 2014, even though the total number of reported colistin results remains unchanged significantly. Our findings are in agreement with previously published research on the trends in P. aeruginosa resistance, which demonstrated an increase in colistin resistance in 2014 ref.52. The potential causes may be associated with the dissemination of resistant strains or laboratory testing methodologies. Nevertheless, we cannot be certain of this without additional molecular data from the isolated isolates. |
|
|
5. While the study confirms established trends, such as high prevalence in ICUs, male predominance, sputum as the primary sample, the authors should highlight unique findings or propose interventions tailored to Saudi Arabia to add new values. |
We are grateful for your invaluable suggestion and agree with it. Consequently, recommendations were provided to mitigate A. baumannii infections based on published research and clinical trials that have been supported by evidence. |
|
|
#- Minor Comments:
|
1-Improve the resolution of Figure 1A and Figure 3 for better readability. |
The issue was addressed and improved, Many thanks for your recommendation, the issue was addressed and improved. |
|
|
2-Address the cutoff issue on the letter "C" in Figure 3. |
Reviewer 4 Report
Comments and Suggestions for Authors
Authors presented the work “Prevalence and Antibiogram Pattern of Acinetobacter baumannii from 2 2013 to 2023 in a Tertiary Care Hospital in the Western Region of Saudi Arabia”.
In general, the work is well-written, it sounds and is relevant. The introduction provides a clear context of the relevance of the pathogen.
Major points of concern:
1. My general impression is that the analysis is very basic, such an epidemiological report, including poor figures in a very summarized way. By considering the journal, I am not sure this work is suitable due a single center is implicated. Inclusion of other centers, overview or measures implemented, and analysis of different molecular results could improve the work, but probably authors don’t have this data.
2. The context provided is very clear. However, after the aim, it may be useful the briefly present the general design of the analysis, for example, that comparison was based on medical services and time, etc.
3. Figure 1: Despite of the title is correct (number of cases), please consider compare (add?) by number of isolates per year -adjusted- or show graphics normalized, and also by number of patients in the hospital per year. A definition of “prevalence”, based on similar studies, may help to avoid bias in the interpretations. For example, during pandemics, does the number of beds increased? Please refer and adjust.
4. The quality of Figures, based on excel graphics, are poorly presented. It applies to Figure 1, 2 and 3. The scale and general presentation is poor. Please re-make!
5. Legends of figures and tables need to be more informative and precise.
6. How are you sure the high rates of resistance to carbapenems are not biased? It is worrisome this rate, considering the international scenario, is it clear that all is ok? Is there no screening to decide which isolates are directed to Antibiotic susceptibility test? Please refer, again, about the definition of prevalence!
Minor points:
· Line 30, Line 100: italics for name. Check in several parts.
· Lines 36-39: Long sentence. Split.
· Line 43: so many references for a very defined statement.
· Line 60: no all antibiotics kill bacteria. Please change the word to clarify and be technically correct (bacteriostatic vrs bactericide).
· Line 55: .. reported many time …. But a single reference is included. Please include more!
· Lines 83-84: references?
· In discussion, it is relevant to mention the role of genomic surveillance (molecular biology, PCR, sequencing, etc), at least in a single paragraph to put the context with other laboratory strategies to monitor resistance and extend this effort.
· Table 5: eliminate “%“ in each number. You already put the “%” in the coliumns title.
· CRAB are define several times.
· “2. Results” are incorrect just before the title of “discussion”
Author Response
Author's Reply to the Review Report (Reviewer 4)
Manuscript number: antibiotics-3436078
Manuscript title: [Prevalence and Antibiogram Pattern of Acinetobacter baumannii from 2013 to 2023 in a Tertiary Care Hospital in the Western Region of Saudi Arabia].
Dear Reviewer, thank you very much for taking the time to review this manuscript. Please find the detailed responses below and the corresponding revisions/corrections highlighted/in track changes in the re-submitted files.
|
Section |
Reviewer comment |
Author response
|
|
#- Major points |
1. My general impression is that the analysis is very basic, such an epidemiological report, including poor figures in a very summarized way. By considering the journal, I am not sure this work is suitable due a single center is implicated. Inclusion of other centers, overview or measures implemented, and analysis of different molecular results could improve the work, but probably authors don’t have this data. |
Thank you for your invaluable recommendations and feedback This study sought to address the gap in understanding the prevalence of A. baumannii in Saudi Arabia. The dataset, which covered a decade, was extensive and required significant effort to isolate A. baumannii from the overall data. Corrections were implemented to include more explanation of statistics and enhance figures of this study were remade in accordance with journal standards. 1. The statistical analysis component of the methodology was revised to include more detailed information regarding database management and the calculation of antibiotic percentages. We also employed ANOVA to compare the means of age groups affected by A. baumannii infections and the means of sample types from which A. baumannii were isolated. 2. Resolution and quality of figures is now improved. As you noted, we lack data from other healthcare centres as well as molecular data regarding the strains. |
|
2. The context provided is very clear. However, after the aim, it may be useful the briefly present the general design of the analysis, for example, that comparison was based on medical services and time, etc.
|
Thank you for this appreciation. - The strategy of this research was based on a study design to evaluate the prevalence of resistance of A. baumannii as a healthcare facility-acquired bacterial pathogen to several antimicrobials over an extended time period, from January 2013 to December 2023.
|
|
|
3. Figure 1: Despite of the title is correct (number of cases), please consider compare (add?) by number of isolates per year -adjusted- or show graphics normalized, and also by number of patients in the hospital per year. A definition of “prevalence”, based on similar studies, may help to avoid bias in the interpretations. For example, during pandemics, does the number of beds increased? Please refer and adjust. |
Thank you for pointing this out. We agree with this comment. Therefore, we have made the correction to address this figure. |
|
|
4. The quality of Figures, based on excel graphics, are poorly presented. It applies to Figure 1, 2 and 3. The scale and general presentation is poor. Please re-make |
Thank you for the constructive comments All figures were improved with more statistical details such as P.value, and CI. GraphPad prism was used for all figures |
|
|
5. Legends of figures and tables need to be more informative and precise. |
The issue was addressed, and more detailed legends were added for all figures and tables, many thanks. |
|
|
6. How are you sure the high rates of resistance to carbapenems are not biased? It is worrisome this rate, considering the international scenario, is it clear that all is ok? Is there no screening to decide which isolates are directed to Antibiotic susceptibility test? Please refer, again, about the definition of prevalence! |
The resistance was comparable to other gram-negative strains reported in published studies from the same geograpgic location. Reference 55, 52, 40 and 42 |
|
|
#- Major points |
1) Line 30, Line 100: italics for name. Check in several parts. 2) Lines 36-39: Long sentence. Split. 3) Line 43: so many references for a very defined statement. 4) Line 60: no all antibiotics kill bacteria. Please change the word to clarify and be technically correct (bacteriostatic vrs bactericide). 5) Line 55: .. reported many time …. But a single reference is included. Please include more! 6) Lines 83-84: references? 7) In discussion, it is relevant to mention the role of genomic surveillance (molecular biology, PCR, sequencing, etc), at least in a single paragraph to put the context with other laboratory strategies to monitor resistance and extend this effort. 8) Table 5: eliminate “% “in each number. You already put the “%” in the coliumns title. 9) CRAB are define several times. 10) “2. Results” are incorrect just before the title of “discussion”
|
Great thanks for pointing these minor corrections: The manuscript is revised, and all minor corrections were addressed, many thanks. |
Reviewer 5 Report
Comments and Suggestions for Authors
In their manuscript, O. Alharbi and colleagues conducted a comprehensive investigation into the prevalence and antibiogram pattern of Acinetobacter baumannii in Saudi Arabia for a period of eleven years.
The strength of this work lies in filling the gap on the A. baumannii infection prevalence and the alterations in antimicrobial resistance. While this manuscript presents a clear introduction and logical flow, there is room for improvements in both the writing and the data presentation, as detailed in the comments below.
1. Figure 1A, what is the color indication? If it means something, please specify. Also, the figure is a bit blurry. Please provide a clear version (at least 300 dpi).
2. Page 3/19, line 118. The reviewer suggests the authors to be consistent with the decimal places. Fore example, if 503.3 is used, then the 21 should be 21.7?
3. Table 1, the reviewer suggests the authors to include a percentage next to the numbers for clear understanding, for example Burn in 2013 is 50 (6.5%).
4. Page 7/19, “In regard to the age, patients in age group of 61-57 exhibits a markedly elevated infection rate relative to other age groups (P<0.0001) as seen in Figure 2.”, the reviewer asks the authors to show the statistical difference in the Figure 2.
5. Figure 2, the reviewer asks the authors to make this figure clear as well (at least 300 dpi) and provide a more detailed caption such as including the error bar and number of replications.
6. Figure 3 was not mentioned in the main text or not cited. Please make sure to cite it in the right place, include y-axis and make the font consistent. The caption needs some changes as well, for example the D was not explained in detail. Is it the comparison between 2013 and 2023 and what is the black dot in the figure.
7. In section 2.3, Figure 4 illustrates a decrease in both the rate and total 5 number of CRAB, dropping from 70% (511 isolates) in 2013 to 60% (340 isolates) in 2023. The reviewer asks the authors to check whether Figure 4 is the right figure.
8. No supporting data or cited figure for “The COVID-19 pandemic did not affect the 8 total number or rate of CRAB infections during 2020 and 2021. The data indicates that the 9 ICUs exhibit the highest incidence of CRAB infections, with reported cases increasing 10 from 21 in 2013 to 150 in 2023.”
9. In general, the reviewer suggests the authors cite the figure when explaining the data and findings in the discussion so the audience will better understand the results and discussion. Also, more discussion on the alterations in antimicrobial resistance should be included.
10. Figure 4, the reviewer did not fully understand this plot.
a. What is the bottom figure and the top figure?
b. Y-axis
c. Caption need more details for understanding.
Comments on the Quality of English Language
Below are some example comments, please check throughout the manuscript again.
1. Page 1, line 27. “Anti-Microbial Resistance Profiles”, no need for capitalization.
2. Figure 1B, be consistent with capitalization.
3. Page 5/19, line 129, leave a space between the ; and the number. For example, (2636; 46%).
4. There is a redundant “2. Results” right before the discussion section.
5. In the discussion, “This rate is consisting ant with”. The “ant” is a typo
6. In the discussion, “trimethoprim/sulfamethoxazole. was reported”. The . is redundant.
Author Response
Author's Reply to the Review Report (Reviewer 5)
Manuscript number: antibiotics-3436078
Manuscript title: [Prevalence and Antibiogram Pattern of Acinetobacter baumannii from 2013 to 2023 in a Tertiary Care Hospital in the Western Region of Saudi Arabia].
Dear Reviewer, thank you very much for taking the time to review this manuscript. Please find the detailed responses below and the corresponding revisions/corrections highlighted in the re-submitted files.
|
Section |
Reviewer comment |
Author response
|
|
|
1. Figure 1A, what is the color indication? If it means something, please specify. Also, the figure is a bit blurry. Please provide a clear version (at least 300 dpi). |
-Thank you for your invaluable recommendations and feedback, all figures quality is now remade to improve quality and readability |
|
2. Page 3/19, line 118. The reviewer suggests the authors to be consistent with the decimal places. Fore example, if 503.3 is used, then the 21 should be 21.7?
|
- Many thanks. A correction has been implemented, and all numerical values have been harmonized.
|
|
|
3. Table 1, the reviewer suggests the authors to include a percentage next to the numbers for clear understanding, for example Burn in 2013 is 50 (6.5%). |
- I appreciate your specific suggestions for revision; the note has been addressed and the entire percentage has been accounted for. |
|
|
4. Page 7/19, “In regard to the age, patients in age group of 61-57 exhibits a markedly elevated infection rate relative to other age groups (P<0.0001) as seen in Figure 2.”, the reviewer asks the authors to show the statistical difference in the Figure 2. |
- In accordance with your comment, we have resolved this matter and incorporated the statistic into the graph, which is further elaborated in the legend. |
|
|
5. Figure 2, the reviewer asks the authors to make this figure clear as well (at least 300 dpi) and provide a more detailed caption such as including the error bar and number of replications. |
-The issue of resolution and caption were addressed and improved, many thanks. |
|
|
|
6. Figure 3 was not mentioned in the main text or not cited. Please make sure to cite it in the right place, include y-axis and make the font consistent. The caption needs some changes as well, for example the D was not explained in detail. Is it the comparison between 2013 and 2023 and what is the black dot in the figure. |
-Many thanks for your comment that we totally agree. For that section 2.3 was added to explain all antibiogram data from table 5 and figure 3 The resistance was comparable to other gram-negative strains reported in published studies from the same geographic location. Reference… |
|
7. In section 2.3, Figure 4 illustrates a decrease in both the rate and total 5 number of CRAB, dropping from 70% (511 isolates) in 2013 to 60% (340 isolates) in 2023. The reviewer asks the authors to check whether Figure 4 is the right figure. |
-Thank you for pointing this issue: According to the data the total A.baumannii isolates were decreased, this might result in reduction in CRAB total isolates. We wanted to show in figure 4 that the reduction in CRAB isolates might related to the overall reduction in total cases. |
|
|
8. No supporting data or cited figure for “The COVID-19 pandemic did not affect the 8 total number or rate of CRAB infections during 2020 and 2021. The data indicates that the 9 ICUs exhibit the highest incidence of CRAB infections, with reported cases increasing 10 from 21 in 2013 to 150 in 2023.” |
-I appreciate your bringing this issue to our attention. The claim is substantiated by the resistance data of imipenem and Meropenem, as the rate remained consistent during the pandemic years, 2020 and 2021. |
|
|
9. In general, the reviewer suggests the authors cite the figure when explaining the data and findings in the discussion so the audience will better understand the results and discussion. Also, more discussion on the alterations in antimicrobial resistance should be included. |
- We are grateful for your recommendation to clarify the text. Therefore, we will make as many references to the table or figure as possible. |
|
|
Quality of English Language
|
10. Figure 4, the reviewer did not fully understand this plot. a. What is the bottom figure and the top figure? b. Y-axis c. Caption need more details for understanding. |
-We are appreciative for your recommendation. The rate (bottom) and counts (top) of CRAB isolates during the surveillance period are depicted in Figure 4. Correction: Detailed explanations of the figure are included in the result and figure legend. |
|
Below are some example comments, please check throughout the manuscript again. 1. Page 1, line 27. “Anti-Microbial Resistance Profiles”, no need for capitalization. 2. Figure 1B, be consistent with capitalization. 3. Page 5/19, line 129, leave a space between the ; and the number. For example, (2636; 46%). 4. There is a redundant “2. Results” right before the discussion section. 5. In the discussion, “This rate is consisting ant with”. The “ant” is a typo
6. In the discussion, trimethoprim/sulfamethoxazole. was reported”. The . is redundant.
|
-Corrections have been made. Many thanks. |
Round 2
Reviewer 3 Report
Comments and Suggestions for Authors
The paper has been adequately improved and is suitable for acceptance.
Reviewer 4 Report
Comments and Suggestions for Authors
Concerns were addressed.
Reviewer 5 Report
Comments and Suggestions for Authors
The reviewer appreciates the authors' effort to improve the manuscript. Despite the enhanced quality, the reviewer have several more comments regarding the figures:
- Figure 2 and 3 are a bit blurry.
- Figure 3 A B C and Figure 4 are missing y-axis label.
- Include the number of replicates and what the error bars indicate in the caption.